# Normalization Bias in Morpho-Transcriptomic Prediction

**Swann Ruyter**[1] iD                                    SWANN.RUYTER@ICM-INSTITUTE.ORG

**Reuben Dorent**[*1] iD                                      REUBEN.DORENT@INRIA.FR

**Daniel Racoceanu**[*1] iD                        DANIEL.RACOCEANU@SORBONNE-UNIVERSITE.FR

[1] *Sorbonne Université, CNRS, Inserm, AP-HP, Inria, Paris Brain Institute - ICM, Paris, France*

## Abstract

Spatial transcriptomics (ST) prediction from H&E histology has attracted growing attention as a scalable approach to infer gene expression from tissue morphology. However, existing methods are typically benchmarked after target normalization, whose impact remains insufficiently characterized. This is critical because raw ST counts partly reflect local cellularity; consequently, normalization determines how much of this morphology-linked variation is retained in the prediction target. Using STEM as a controlled diffusion-based backbone, we compare its original log-normalization with a Pearson-residual-based normalization on a HER2-positive breast cancer ST cohort. We show that log-normalization preserves substantially stronger nuclei- and depth-related signals, and that the best-predicted genes under this regime are also the most strongly associated with nuclei-related features in the observed data. These findings suggest that part of the apparent performance gain in log-transformed space arises from the preservation of an easier-to-predict cellularity-related signal, with important implications for method comparison and for the interpretation of correlation-based benchmarks.

**Keywords:** Spatial transcriptomics, diffusion models, normalization, H&E.

## 1. Introduction

Spatial transcriptomics (ST) enables *in situ* measurement of gene expression while preserving tissue architecture, thereby providing a direct link between routine H&E histology and spatially resolved molecular states (Rao et al., 2021). Given the widespread availability of H&E slides and the high cost and limited scalability of ST, recent work has focused on morpho-transcriptomic approaches that infer gene expression directly from tissue morphology (e.g.,Pang et al., 2021; He et al., 2020; Zeng et al., 2022; Zhu et al., 2025).

Current methods are mostly evaluated with correlation- or error-based metrics, such as Pearson correlation coefficient (PCC) and root mean squared error (RMSE), computed on normalized transcriptomic targets. However, the effect of target normalization itself is rarely isolated (Wang et al., 2025). This issue is particularly critical in ST, where sequencing depth directly affects the stability of expression estimates, while also depending in part on the amount of biological material captured within each spot, and thus on local tissue cellularity.

As a result, target normalization can either preserve or suppress morphology-associated signals linked to cellularity and capture efficiency, thereby introducing a potential bias in the evaluation of predictive models.

In this work, we use STEM (Zhu et al., 2025) to compare the effect of its original log-transformed target with a Pearson-residual-based normalization (Lause et al., 2021). We

---

\* Contributed equally

show that the log-transformed target retains a stronger cellularity- and sequencing-depth-related axis, and that its best-predicted genes are also the most strongly associated with nuclei-related features. These findings suggest that part of the apparent correlation gain in log-transformed space arise from the preservation of easier-to-predict cellularity-related variation, potentially confounding method comparison and biasing the assessment of finer morpho-transcriptomic signals.

## 2. Method

We use STEM (Zhu et al., 2025) as a controlled morpho-transcriptomic backbone, in which a conditional denoising diffusion model generates spot-level transcriptomic predictions from corresponding histological patch embeddings. Local cellularity is estimated from H&E images via nuclei segmentation using TIAToolbox (Pocock et al., 2022). Nuclei counts are aggregated at the spot level and used as a proxy for captured biological material.

To quantify the effect of target normalization, we compare STEM's original log-transformed normalization (Eq. (1)) with a Pearson-residual-based normalization (Eq. (2)) under identical *one-slide-out* predictions and matched observed targets. We then derived two complementary analysis families: (i) coupling between observed targets and nuclei count, and (ii) nuclei association among the top-ranked predicted genes.

## 3. Experiments

We evaluate our approach on the public ST cohort HER2-positive breast cancer (Andersson et al., 2021), harmonized in HEST-1K (Jaume et al., 2024) (35 slides after QC). All experiments follow a *one-slide-out* protocol, ensuring strict separation between training and test slides. A *one-slide-out* protocol ensures strict separation between training and test data. A panel of 300 genes is selected by combining per-slide variability with global ranking by expression and dispersion. All experimental settings are fixed, and only the target normalization is varied.

Figure 1 panels A–C, reveal that the effect is already present in the observed data. Nuclear density correlates with sequencing depth across slides, with a median slide-level Spearman correlation of 0.505, indicating that nuclei count captures a meaningful proxy for the amount of material captured per spot. This dependence is substantially better preserved in the log-transformed target than in the Pearson-residual target, both in terms of median gene-wise nuclei association (panel B; median $|\rho(\text{true log expr}, n_{\text{nuclei}})| = 0.337$ vs. $0.085$) and the proportion of nuclei-associated genes (panel C; 0.68 vs. 0.03 for $|\rho| > 0.3$). These results show that normalization affects both scaling and the amount of retained cellularity-related signal (Supplementary Table 1).

Figure 1 panels D–F shows that the apparent performance advantage of log-space prediction is not uniform across genes. Although log-transformed targets yield higher predictive correlations for the top-ranked genes (panel D), these genes are also more strongly associated with nuclear density in the observed data than the corresponding top-ranked genes in Pearson space (panel E). This effect is stable across all the top-$k$ thresholds considered, with a median paired excess in nuclei association ranging from 0.257 (top-10) to 0.273 (top-200). All paired Wilcoxon tests are significant ($p \leq 2.9 \times 10^{-7}$) and permuta-

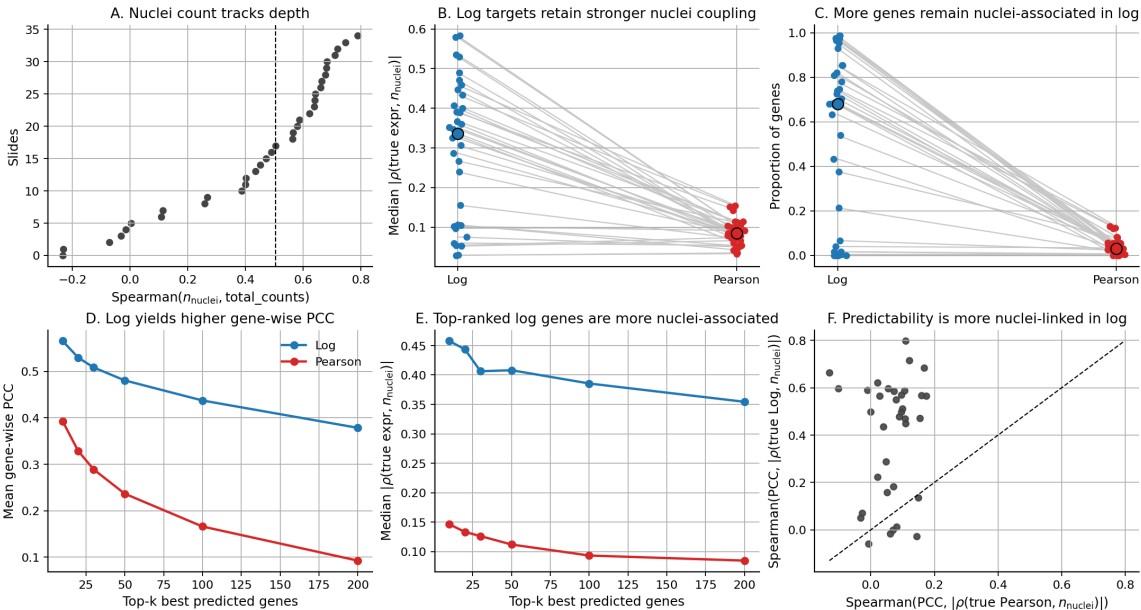

Figure 1: **Target normalization alters observed signal and apparent predictive gain. A–C**: Log targets retain stronger cellularity-related signal than Pearson targets. **D–F**: Higher correlations in log space concentrate on genes more associated with nuclei count.

tion tests were consistent ($p \leq 2.0 \times 10^{-4}$). The same trend holds at slide level, where gene predictability is more tightly coupled to nuclei dependence in log-transformed space (panel F). Together, these results indicate that part of the correlation gain in log space reflects preserved cellularity-related variation rather than finer morpho-transcriptomic recovery (Supplementary Table 2).

## 4. Conclusion

In summary, transcriptomic target normalization changes not only what is predicted, but also how predictive performance should be interpreted. In our setting, log-normalized targets preserve a stronger cellularity-related signal than Pearson-residual-normalized targets, and this signal is concentrated among the genes that appear best predicted. As a result, correlation-based benchmarks may partly reward coarse, easier-to-predict cellularity-related variation rather than finer morpho-transcriptomic recovery, depending on the biological question of interest. Although this study is limited to a single backbone and cohort, it demontrates that normalization should be treated as a central experimental factor rather than a secondary preprocessing choice.

## Acknowledgments

The research leading to these results has received funding from Agence Nationale de la Recherche as part of the "France 2030" program (reference ANR-23-IACL-0008, PRAIRIE-PSAI) and as part of the "Investissements d'avenir" program (reference ANR-19-P3IA-0001, PRAIRIE 3IA Institute; and reference ANR-10-IAIHU-0006). The ARAMIS Lab is affiliated with DIM C-BRAINS, funded by the Conseil Régional d'Ile-de-France. This work was performed using HPC resources from GENCI–IDRIS (Grant 2025-AD011016416). R.D. received a Marie Skłodowska-Curie grant No 101154248 (project: SafeREG).

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

## Supplementary Material

### Logarithmic normalization

For the logarithmic target used in the original STEM setting, raw counts are transformed element-wise without library-size normalization:

$$Y_{ij}^{\log} = \log_2(X_{ij} + 1). \tag{1}$$

This representation compresses the dynamic range of count values but does not explicitly correct for sequencing-depth effects.

### Pearson Residual Normalization

Let $X \in \mathbb{R}_{\geq 0}^{N \times G}$ be a raw count matrix, where $N$ and $G$ are respectively the number of spots and genes. We compute Pearson residuals under a negative binomial (NB) model with fixed dispersion $\theta$. First, we define the library size $n_i = \sum_j X_{ij}$, the gene proportion $p_j = \frac{\sum_i X_{ij}}{\sum_{i,j} X_{ij}}$, and the expected mean $\mu_{ij} = n_i p_j$. Then, Pearson residuals are computed as

$$R_{ij} = \frac{X_{ij} - \mu_{ij}}{\sqrt{\mu_{ij} + \mu_{ij}^2/\theta}}, \tag{2}$$

where $\theta$ denotes a fixed dispersion parameter and a numerical floor of $10^{-8}$ is applied to the denominator for stability. This formulation leads to approximately variance-stabilized residuals across genes and expression levels.

### Supplementary Statistical Tables

Table 1: **Observed targets retain different amounts of nuclei- and depth-related signal.** Slide-level summary statistics computed from the observed transcriptomic targets. Strong nuclei association is defined as $|\rho(\text{expression}, n_{\text{nuclei}})| > 0.3$.

| Metric | Log | Pearson |
|---|---|---|
| Median $|\rho(\text{expr}, n_{\text{nuclei}})|$ | 0.337 | 0.085 |
| Median proportion with $|\rho(\text{expr}, n_{\text{nuclei}})| > 0.3$ | 0.680 | 0.030 |
| **Depth coupling** | | |
| Median Spearman$(n_{\text{nuclei}}, \text{total\_counts})$ | | 0.505 |

Table 2: **Top-$k$ best-predicted genes are more nuclei-associated in log space.** For each $k$, we report the slide-level median paired difference $\Delta_k = |\rho(\text{true log}, n_{\text{nuclei}})| - |\rho(\text{true Pearson}, n_{\text{nuclei}})|$ computed on the top-$k$ best-predicted genes, together with paired Wilcoxon and permutation tests.

| Top-$k$ | Median $\Delta_k$ | Wilcoxon $p$ | Permutation $p$ |
| --- | --- | --- | --- |
| 10 | 0.257 | $2.9 \times 10^{-7}$ | $2.0 \times 10^{-4}$ |
| 20 | 0.266 | $7.1 \times 10^{-7}$ | $1.0 \times 10^{-4}$ |
| 50 | 0.269 | $6.9 \times 10^{-8}$ | $1.0 \times 10^{-4}$ |
| 200 | 0.273 | $6.0 \times 10^{-9}$ | $1.0 \times 10^{-4}$ |

