# OpenReview forum: "Normalization Bias in Morpho-Transcriptomic Prediction"
_MIDL.io/2026/Short_Papers — MIDL 2026 - Short Papers Poster_

### Official Review · Reviewer_J7An · 2026-05-03
**Normalisation bias in morpho-transcriptomic prediction from H&E**

**Rating:** 5
**Confidence:** 4

**Review:**

This paper makes a focused and well-argued methodological contribution that is directly relevant to the growing field of spatial transcriptomics prediction from histology. The core finding—that log-normalisation preserves a cellularity-related signal that can inflate apparent predictive performance—is important, timely, and supported by rigorous statistical analysis. While the study is limited to a single backbone and cohort, this is appropriate for a short paper and is clearly acknowledged by the authors. Overall, this work is of strong interest to the MIDL community.

**Summary:**

This short paper investigates the effect of target normalisation on the evaluation of spatial transcriptomics (ST) prediction from H&E images. Using a diffusion-based backbone (STEM), the authors compare log-transformed targets with Pearson-residual normalisation on a HER2-positive breast cancer spatial transcriptomics cohort. They show that log-normalization preserves stronger cellularity- and sequencing-depth-related signals, which are easier to predict from morphology and drive higher correlation scores. The work highlights normalisation as a critical and often overlooked factor in benchmarking morpho-transcriptomic models.

**Strengths:**

- Introduces a clear and impactful conceptual insight: normalisation can bias morpho-transcriptomic benchmarks
- Clean and well-controlled experimental design
- Empirical evidence supported by statistical testing
- The finding that performance gains concentrate on genes already strongly associated with nuclei effectively disentangles normalisation bias from true predictive signal.

**Weaknesses:**

- The paper relies entirely on STEM as its backbone, but the description is minimal. It is okay for a short paper, but it hinders readability.
- The study is limited to one model and one dataset. While acknowledged, it remains unclear how well the observed bias generalizes across architectures or tissue types.

**Justification Of Rating:**

This short paper presents a clear and impactful insight: target normalisation can bias morpho-transcriptomic benchmarks by preserving easier-to-predict cellularity-related signals, inflating apparent performance. The point is timely, directly relevant to ST prediction from H&E, and supported by statistical analysis. While limited to a single model and cohort, this is appropriate for a short paper and acknowledged by the authors. Overall, it offers valuable critical evaluation that is of strong interest to the MIDL community.

---

### Decision · Program_Chairs · 2026-05-08

Accept (Poster)